# People are surprisingly hesitant to reach out to old friends
Lara B. Aknin ⓘ [1,3] ✉ & Gillian M. Sandstrom ⓘ [2,3]

Social relationships provide one of the most reliable paths to happiness, but relationships can fade for various reasons. While it does not take much to reinitiate contact, here we find that people are surprisingly reluctant to do so. Specifically, most people reported losing touch with an old friend yet expressed little interest in reaching out (Studies 1-2, $N$s = 401 and 199). Moreover, fewer than one third of participants sent a message to an old friend, even when they wanted to, thought the friend would be appreciative, had the friend's contact information, and were given time to draft and send a message (Studies 3-4, $N$s = 453 and 604). One reason for this reluctance may be that old friends feel like strangers. Supporting this possibility, participants were no more willing to reach out to an old friend than they were to talk to a stranger (Study 5, $N$ = 288), and were less willing to contact old friends who felt more like strangers (Study 6, $N$ = 319). Therefore, in Study 7 ($N$ = 194), we adapted an intervention shown to ease anxieties about talking to strangers and found that it increased the number of people who reached out to an old friend by two-thirds.

Evidence from across the social sciences demonstrates that social relationships provide one of the most robust and reliable routes to well-being. For instance, individuals with strong and satisfactory relationships report the highest levels of happiness[1,2], and people who have someone to count on in times of need report higher life evaluations worldwide[3].

While the quality of relationships matters, so too do the quantity and diversity of social connections. Social network size is positively associated with greater well-being[4,5] and recent work spanning multiple international data sets indicates that people who have more diverse relationship networks also report greater well-being[6]. These findings align with recent theorizing in relationship science which cautions against relying on any one person to fulfill all of one's emotional needs[7]. Instead, people who turn to different social connections for different emotion regulation needs (e.g., calling on one person to cheer them up when they are sad, and a different person to calm them down when they are anxious) report higher well-being[8]. Thus, although classic work indicates that high-quality relationships are necessary for happiness[1,9], recent research suggests that having more diverse relationships is also a predictor of well-being.

People recognize that relationships are an important source of personal meaning and well-being[10,11], yet life can get busy and compel various relationships to fade or be put on hold. The high priority placed on work and productivity in North America has led people to cut back on social connections and social time to meet increasing demands at work[12,13]. Indeed, a significant majority of working Americans feel as if they do not have enough time in the day[14]. Social withdrawal may also occur in more discrete episodes, such as when people navigate life transitions to parenthood or a new job, and contributes to elevated feelings of loneliness during these pivotal times[15–17].

While the strength of friendships may naturally wax and wane[18], neglecting relationships for too long can be problematic. Loneliness is defined as a perceived lack of social connection, and it predicts a range of mental and physical health challenges[19,20]. Given the clear importance of social connection, Sociometer Theory[21] posits that self esteem functions as a psychological gauge to indicate the extent to which one feels accepted and socially valued. This gauge alerts people when social connection levels decline too far, and compels people to prioritize and strengthen social relationships. But how is one to do so?

Reaching out to an old friend with whom one has lost touch offers one accessible and viable channel for bolstering and diversifying social connection. For instance, a person could visit, call, email or send a text message to a friend, colleague, or family member that they like and care about but have not seen in some time (which we refer to as an "old friend"). Such efforts to reconnect are likely more efficient than initiating a new friendship; research estimates that it takes more than 200 hours of contact to turn a new acquaintance into a close friend[22]. This may be why empirically-informed programs, such as Groups4Health, recommend that individuals who are lonely consider reconnecting with old friends[23]. Moreover, research suggests that reaching out to an old friend can be beneficial. One study that asked

[1]Department of Psychology, Simon Fraser University, Burnaby, BC, Canada. [2]Department of Psychology, University of Sussex, Brighton, UK. [3]These authors contributed equally: Lara B. Aknin, Gillian M. Sandstrom. ✉e-mail: lara_aknin@sfu.ca

MBA students to solicit help or advice on a work project found that reconnecting with "dormant ties" provided more useful knowledge and insight than connecting with current strong ties[24].

While reaching out to old friends may be practical, this strategy may not be enacted because various psychological hurdles hinder people's willingness to connect with others[25]. Indeed, recent work indicates that people overestimate the awkwardness of reaching out to an old friend and underestimate the appreciation and positive feelings such acts foster[26–28]. In addition, people misestimate the outcomes of other social acts involving other social partners. For instance, although talking to strangers can boost short-term happiness, people expect the opposite[29,30]. Similarly, people systematically overestimate how uncomfortable it will be to express gratitude and fail to recognize how much a compliment means to the recipient[31–33]. Collectively these findings indicate that people hold a number of faulty assumptions about the realities and consequences of various social interactions.

Critically, systematically underestimating others' appreciation for one's social behaviour (e.g., reaching out, talking to a stranger, giving a compliment) is expected to make people more reluctant to engage in these activities because they are missing the full motivation to act. While this premise is grounded in common sense and psychological theory[34], additional research is needed to test the extent to which these misestimations translate into refraining from engaging in the behaviour. One study attempted to increase the number of people reaching out to an old friend by teaching participants about misestimation errors[28]. Unfortunately, this intervention did not translate into more people actually reaching out to an old friend. Thus, while past work has demonstrated that people systematically underappreciate how much social targets value interactions, including being contacted by an old friend, here we explore people's self-reported, and actual willingness to engage in these actions, as well as how to promote this valuable behaviour.

Are people reluctant to reach out to old friends, why might this be, and how can they be encouraged to reconnect? We examine these questions in seven studies. In Study 1, we ask what proportion of people have lost touch with an old friend, how willing they are to reach out, what barriers restrain them, and what reasons would encourage them to reach out. After observing a general reluctance to reach out, in Study 2 we investigate whether people are hesitant about the idea of reconnecting with an old friend or simply aversive to the idea of being the one to reach out. Then, in Studies 3 and 4, we test multiple interventions designed to address some of the barriers identified in Study 1. These efforts have little influence on the proportion of people who actually reach out to an old friend when given the opportunity to do so.

In light of these data, we reasoned that one explanation for why people may be reluctant to reach out to old friends is because old friends may feel like strangers once substantial time has passed. Consistent with this possibility, several of the barriers that participants endorsed when thinking about reaching out to old friends are similar to the barriers that make people reluctant to talk to strangers. Therefore, in Study 5 we benchmark people's willingness to reconnect with an old friend against several daily behaviours, including talking to a stranger. In finding some evidence that reaching out to an old friend may be psychologically similar to talking to a stranger, in Study 6 we examine whether people are more reluctant to reach out to old friends when those friends feel more like strangers. Finally, in Study 7, we take lessons from an intervention that has lastingly eased anxieties about talking to strangers. By applying a similar design in which we target participants' behaviour rather than their attitudes, we effectively encourage more people to reach out to old friends. Two additional studies are included in the Supplementary Information (SI): Supplementary Note Study S8; Supplementary Note Study S9.

## Methods

Studies 1, 5, and S8-9 were approved by the Office of Research Ethics at Simon Fraser University (application numbers: 30000726, 30001307, 30001308, and 30002053, respectively). Studies 2, 3, 4, 6, and 7 were approved by the Sciences & Technology Cross-Schools Research Ethics Committee at the University of Sussex (application numbers: ER/GS474/1, ER/GS474/3, ER/GS474/6, ER/GS474/9, and ER/GS474/10, respectively). All participants provided informed consent before participation and studies were conducted in compliance with all relevant ethical guidelines. Studies 2–7 and both supplemental studies in the Supporting Information were pre-registered. Pre-registration links and the dates they were posted are as follows: Study 2 (osf.io/93mwh; April 13, 2022), Study 3 (osf.io/ynt63; July 20, 2022), Study 4 (osf.io/npwa4; December 11, 2022), Study 5 (osf.io/bm3x7; February 10, 2023), Study 6 (osf.io/phrc9; October 16, 2023), Study 7 (osf.io/rzpu8; November 14, 2023), Study S8 in SI (aspredicted.org/q4gj8.pdf; September 13, 2022), and Study S9 in SI (aspredicted.org/bq7cs.pdf; October 3, 2023). We deviated from the Study 2 pre-registration in that we had originally planned to recruit 200 young adults and 200 older adults for this study, but ultimately only young adults participated. For clarity and transparency, we report the results of all pre-registered hypotheses in the main text. In the Methods section, we fully describe the measures that correspond to the results that are reported in the main text, and then name any additional variables that were not analyzed, which can be viewed in the materials on OSF.

All data were collected on Qualtrics, and random assignment to condition (i.e., in Studies 2, 3, 4, and 7) was done by Qualtrics. Participants self-reported their gender in each study. In all studies, data distributions were assumed to be normal but this was not formally tested. All samples were convenience samples, except for Study S8 in which we collected data from a nationally representative sample of Americans.

### Study 1

**Participants.** Four-hundred forty-one undergraduates at a university in Canada participated as part of a larger study in exchange for course credit. Of these, 40 (9%) had never lost touch with someone, and were not invited to continue with the survey. This left a final sample of 401 participants ($M_{age}$ = 19.2, $SD$ = 2.0; 305 women, 86 men, 10 other). Sample size calculations were conducted a priori for a separate research question.

**Procedure.** Participants completed an online survey in a private room. After answering several questions about an unrelated topic, they were asked to indicate whether they had "lost touch with a friend that [they] care about." If yes, participants were asked to provide their old friend's initials to personalize the following questions. If no, participants did not complete the remaining questions, and were not included in the analyses.

**Measures.** Participants were asked how willing they would be to reach out to their old friend via phone, text, or email to say hello, both in general (i.e., with no timeframe specified) and right now, on a Likert scale with anchor labels: 1 = not at all, 4 = neutral/undecided, 7 = extremely.

Because expectations about the recipient's response are likely to impact how willing someone may be to reach out[28,35,36], we asked participants two questions about how positively their friend would evaluate them and their message if they were to reach out (1 = very negative to 7 = very positive).

We asked participants to what extent each of the following barriers held them back from reconnecting with the friend in question or other friends they have not been in touch with for a while, using a 7-point scale (1 = not at all relevant, 7 = extremely relevant). The potential barriers were: (i) my time is limited, (ii) their time is limited, (iii) I don't have time for a longer catch-up right now, (iv) I don't have anything important to say, (v) I'm not sure I'll get the wording just right, (vi) it would be awkward to reach out after all this time, (vii) I don't know if they are interested in hearing from me, and (viii) I don't want to bother them. Participants could also type another reason for not reaching out, if desired.

Finally, we asked participants to what extent they would be willing to reach out to their old friend for each of the following reasons (1 = not at all, 7 = extremely): (i) your friend's birthday, (ii) a holiday (e.g., New Year's), (iii) because something reminded you of a shared experience, (iv) just because

(no particular reason), (v) you were thinking about them, (vi) you heard a good joke, saw a cute picture/video, or thought of something they might enjoy, (vii) you were going to be in their neighborhood, or near their workplace, and (viii) to ask for help/advice. Participants could also type another reason for reaching out, if desired.

Several additional measures were included and are not reported in the main text. For instance, we asked participants how they would feel if they didn't reach out, and the extent to which they and their old friend would view reaching out as an act of kindness.

## Study 2

**Participants**. A total of 266 young adults from the United Kingdom and United States, recruited on Prolific in exchange for payment, answered questions as part of a larger study. This sample size was calculated a priori to provide appropriate power for the larger study. Of those, 67 (25%) had never lost touch with someone, and were not invited to continue with the survey. This left a final sample of 199 participants ($M_{age}$ = 27.4, $SD$ = 1.9; 122 women, 73 men, 4 other; $n_{UK}$ = 94, $n_{US}$ = 58, $n_{missing}$ = 47).

**Procedure**. As in Study 1, participants were asked to indicate whether they had lost touch with a friend they care about and, if so, to provide their old friend's initials. Participants were then randomly assigned to think about either *reaching out to* ($n$ = 100) or *hearing from* ($n$ = 99) the old friend.

**Measures**. Using similar questions as in Study 1, participants were asked how interested they would be to reach out to [hear from] their old friend via phone, text, or email to say hello—sometime in the future and right now. Responses were provided on scales ranging from 1 = *not at all*, to 7 = *definitely*.

Once again, several additional measures were included, and are not reported in the main text. As in Study 1, we asked participants to what extent various barriers held them back from reaching out, and how willing they would be to reach out given various reasons (see the Supplementary Note, Study 2, including Supplementary Figs. 1, 2, for results related to these measures). We also asked participants how positive/negative they would feel if they reached out to/heard from their old friend, and how positive/negative they would feel if they/their friend wanted to reach out but decided not to. Finally, we asked participants the extent to which they consider reaching out to/hearing from their old friend as an act of kindness.

In addition to the central prediction that participants would be more interested in hearing from than reaching out to an old friend, we predicted that people would see each reason (e.g., because it's their birthday) as better justification for hearing from vs. reaching out to an old friend. We report the results of this hypothesis in the Supplementary Note, Study 2.

## Study 3

**Participants**. A total of 495 participants from the United Kingdom, the United States and Canada started this experiment on Prolific in exchange for payment. Of those, 28 had never lost touch with someone (i.e., they did not pass our screening question), and 14 chose not to continue with the full study when given the option after completing this screening question. This left a final sample of 453 people ($M_{age}$ = 39.3, $SD$ = 12.8; 237 women, 213 men, 3 other; $n_{UK}$ = 334, $n_{US}$ = 90, $n_{Canada}$ = 29). A priori calculations indicated that a sample of 432 was needed to detect a small/medium effect ($f$ = 0.15) with 80% power, using a between-subjects ANOVA with alpha set to .05.

**Procedure**. At the start of the study, participants were asked to indicate whether they had lost touch with someone who (i) they would be happy to reconnect with, (ii) they had contact information for, and (iii) they thought would like to hear from them. Only participants who were able to identify a target meeting these criteria were allowed to proceed to the study, where they were asked to provide the initials for the person so that the remaining questions could be personalized.

Participants were then asked to imagine that they were going to reach out to the person they had identified, and were given 2 min to draft a "hello" message. They were told that the message could be as short or long as they wanted. Participants were informed that they could not proceed until the 2 min had passed, so they should use the time to type a short message. All participants chose to type a message, such as "Hello you. It's been an age again. Hope you've been keeping well, Miss you." Messages ranged from 3 words ("hello, hows life") to 184 words ($M$ = 42.6, $SD$ = 25.7).

**Manipulation**. Participants were randomly assigned to see one of three prompts encouraging them to send their note.

In the *control condition* ($n$ = 149), the prompt said: "We encourage you to take this time to open an email or text message and send the message you wrote."

In the *reflective condition* ($n$ = 151), the following was added to the control prompt: "Think about how much you would appreciate it if you got a note from [your old friend]. Someone has to reach out first - why not you?"

In the *impulsive condition* ($n$ = 153), the following was added to the control prompt: "If you are having second thoughts, we suggest you do not entertain them. Don't doubt yourself - just open an email or text message, paste in your note, and press 'send.'"

The two interventions were intended to reflect the rich history of dual processing models in psychology, which suggest that people have two thinking styles: one that is slower, more effortful, deliberate, and reflective, and another that is faster, more effortless, impulsive, and intuitive[37–40]. Participants in all conditions were told that they could not proceed in the survey until 1 min had passed.

**Measures**. Our pre-registered dependent variable was whether participants sent a message to their old friend. To capture this behaviour, we asked participants whether they sent the message, and provided three response options: yes, no, and "maybe later", which was included to encourage honesty. We also encouraged honesty by assuring participants that their pay would not be impacted by their response. We pre-registered our intention to treat "maybe later" as "no," because we wanted to measure actual behaviour, rather than intentions.

After deciding whether or not to send a message to their old friend, participants reported their current positive ($a$ = 0.93) and negative emotion ($a$ = 0.90) on the Positive and Negative Affect Schedule[41], with "happy" added as an additional positive emotion item.

Again, several additional measures were included and are not reported in the main text. We asked participants how much they considered the following while making their decision: possible rewards (to be nice, because they miss their old friend), and a range of barriers (similar to the ones in Study 1), including worries about potential attributions that their old friend might make (that they were lonely or had an ulterior motive; see the Supplementary Note, Study 3, including Supplementary Figs. 3, 4, for results related to these measures). We asked people who chose to reach out to describe their biggest motivator, and we asked people who chose not to reach out to describe their biggest barrier. Finally, we asked participants the extent to which their old friend would consider their message an act of kindness (or would have done so if they had chosen to reach out).

## Study 4

**Participants**. A total of 732 participants from the United Kingdom, United States, and Canada started this experiment on Prolific in exchange for payment. Of those, 63 had never lost touch with someone (i.e., they did not pass our screening question), 55 thought of someone who did not meet our eligibility criteria (see below), and 10 chose not to continue to the full study when given the option, after completing this screening question and thinking of someone who met all the eligibility criteria. This left a final sample of 604 people ($M_{age}$ = 40.5, $SD$ = 13.0; 274 women, 327 men, 3 other; $n_{UK}$ = 455, $n_{US}$ = 128, $n_{Canada}$ = 21), which surpassed our target sample of 600 participants calculated a priori to provide 90% power to detect a small/medium effect ($f$ = 0.15) with alpha set to 0.05.

**Procedure**. Participants were asked to indicate whether they had lost touch with someone, using the same instructions as in Study 3, but with the addition of specifying that it should be someone they had lost touch with for no particular reason (i.e., not a falling out). To confirm eligibility, we asked people to tick a box to indicate that the person they were thinking of met each of our criteria: (i) someone who they would be happy to reconnect with, (ii) someone they had lost touch with for no particular reason, (iii) someone for whom they had contact information at hand, and (iv) someone they thought would like to hear from them. If they ticked all four boxes, they were able to continue the survey; if they did not tick even one of the boxes, they were not invited to continue to the full survey.

**Manipulation**. Participants were randomly assigned to one of three conditions. In the *message condition* (which was similar to the control condition in Study 3; *n* = 204), participants were given 2 min to compose a short message to their old friend, and were not able to proceed until 2 min had elapsed. Afterwards, participants were given 1 min to send their message. Participants saw a prompt saying, "Now we'd like to give you the opportunity to reach out to [friend's initials]. We encourage you to take this time to open an email or text message and send a message." The note the participant had written was shown on the screen so that participants could copy and paste it into a message if they wanted.

Participants in the *message plus encouragement condition* (*n* = 206) received the same instructions as the *message condition* but were additionally told: "Research suggests that sending a short message to someone to say that you are thinking of them (or hope they are well) is an act of kindness —and that this gesture is likely to be appreciated by your friend, even more than you expect. Also, a note of this sort does not suggest to your friend that you expect a response or require any further contact, so your message has low potential for risk, and high potential for reward." We thought this intervention would (i) position the hello message as an act of kindness, to minimize concerns that it was a bother for the recipient, and (ii) reduce fears of rejection, by suggesting that participants should not expect a reply. As in the message condition, the note the participant had written could be copied and pasted into a message.

Finally, in the *control condition* (*n* = 194), participants were not given time to prepare a note, but were instead given 2 min to write about a typical day, and were not able to proceed until 2 min had elapsed. Afterward, participants saw a prompt saying: "Now we'd like to give you the opportunity to reach out to [friend's initials]. We encourage you to take this time to open an email or text message and send a message."

Messages written by participants in the experimental conditions ranged from 1 word ("hello") to 129 words (*M* = 44.2, *SD* = 21.4).

**Measures**. We asked participants if they sent their message to their old friend or not, using the same question used in Study 3. Again, we pre-registered our intention to treat "maybe later" as "no." We also asked participants how much they had considered several barriers while making their decision (see Supplementary Note, Study 4 for results). We used a shorter list of barriers than in earlier studies, including only the ones that we thought might be affected by the manipulation.

## Study 5
**Participants**. A total of 303 participants were recruited in public spaces on a university campus in Canada in exchange for candy. As required by the local ethics board at the site of data collection, participants were asked before the study to provide informed consent for participation and, separately, to grant permission to share their responses in an online repository for open science initiatives. We report results from the sample of 288 participants who gave permission to share their data (*M*_age = 20.7, *SD* = 2.9; 172 women, 107 men, 5 gender fluid/non-binary/both, 4 participants with undisclosed gender), so that these findings can be replicated with the file posted on the OSF. Findings do not differ in the full sample. We pre-registered our intention to recruit at least 275

participants to provide 90% power to detect a small size effect (*d* = 0.2) with a paired samples t-test and alpha at 0.05.

**Procedure**. Participants completed a short online survey in which they were asked to rate their willingness to engage in various common activities right away. To increase the believability that participants may be asked to complete a task immediately, we kept props for some actions nearby, including a cooler bag to hold ice cream bars, bags of coins, a hand grip, and a large garbage bag for trash collection.

**Measures**. Participants rated their willingness to complete eight everyday activities right now on a scale ranging from 1 = *extremely unwilling* to 7 = *extremely willing*: (i) call or text an old friend that you have lost touch with, (ii) talk to a stranger, (iii) listen to a song you loved in your childhood or teen years, (iv) eat an ice cream bar, (v) sort a bag of coins, (vi) hold a hand grip for 30 s, (vii) book a dentist appointment or physical exam, and (viii) pick up litter. Items were presented in random order.

Participants were asked whether they had lost touch with a friend they care about (yes/no), and whether they had ever thought about reaching out but did not (yes/no). See Supplementary Note, Study 5 for detailed results on these exploratory measures.

## Study 6
**Participants**. A total of 505 participants were recruited from the United Kingdom, the United States and Canada on Prolific in exchange for payment. They completed a pre-screening survey to see if they could identify three to five people they "haven't been in touch with for a while." Of these, 502 were able to do so, and were invited to complete the full survey, though we limited participation to 320 people. Of the 324 participants who completed the full survey, 319 (*M*_age = 39.5, *SD* = 13.4; 138 women, 176 men, 5 other; *n*_UK = 171, *n*_US = 118, *n*_Canada = 30) passed our pre-registered attention check and form our final sample.

Given the challenges involved in power analysis for mixed models, we based our power analysis on a between-subjects design, which should be more conservative. Our power analysis suggested that, in order to have 80% power to detect a small sized bivariate correlation (*r* = 0.15) with an alpha of 0.05, we needed 273 participants, so we pre-registered a recruitment target of 300 participants.

**Procedure**. Participants completed a short online survey in which they named three to five people they had not been in touch with for a while ("old friends"; *n* = 121 people named three old friends, *n* = 55 named four, and *n* = 143 named five), and answered a few questions about each old friend, including familiarity, and willingness to reach out. For exploratory purposes, participants also named a current friend (someone they "know fairly well and have recently been in touch with") and a new acquaintance ("someone [they] recently met and interacted with for the first time"), and answered the same questions about these targets (see Supplementary Note, Study 6 for analyses involving these targets).

**Measures**. Participants indicated the type of relationship they had with each target, by ticking all that apply from a list of seven options (or "other").

Participants reported how recently they had been in touch with each target, on a 5-point scale from 1 = *more than a few months ago* to 5 = *in the last few days*. Critically, participants rated how well they currently know each target, on a 7-point scale from 1 = *I know them as well as a stranger* to 7 = *I know them as well as I know myself*. Finally, participants rated their willingness to reach out right now (via phone, text, email, social media, or in-person) to say hello to each target, on a 7-point scale from 1 = *very unwilling* to 7 = *very willing*.

## Study 7
**Participants**. A total of 348 people were recruited in person on a university campus in the U.K. and were reimbursed with chocolate and a

chance of winning a draw prize. Of these, 237 were eligible to complete the survey, because they were able to think of someone they had lost touch with who met all of our criteria - the same criteria used in Study 4, which we verified using tick boxes, as in Study 4. We excluded two additional participants because they were taking a class taught by one of the authors, in which some of the studies in the current paper had been discussed. Our final sample consisted of 194 participants ($M_{age}$ = 23.2, $SD$ = 7.5; 112 women, 65 men, 10 other ways, and 7 participants with undisclosed gender) who answered the key question about whether or not they had reached out to their old friend. This final sample surpasses our pre-registered target sample of 160 participants needed to provide 80% power to detect a medium size effect ($dz$ = 0.4) with an independent samples t-test and a one-tailed alpha of 0.05.

**Procedure.** Participants completed an online survey in which they thought of someone they had lost touch with. They were randomly assigned to either the practice ($n$ = 101) or no-practice (i.e., control; $n$ = 93) condition, in each of which they completed a task for 3 min. Participants in the practice condition were asked to "send messages (via text, chat, etc.) to several current friends/acquaintances", and on average they sent messages to about three people ($M$ = 3.3, $SD$ = 1.8). Participants in the no-practice condition were asked to "browse several social media accounts/feeds", and on average they browsed six or seven accounts ($M$ = 6.7, $SD$ = 14.5). Participants were not able to continue to the next page of the survey until 3 min had elapsed. Next, participants in both conditions were encouraged to send a message to their old friend, and were told that it was an act of kindness that would benefit them (increase their happiness), and would be appreciated by their friend. Participants were not able to continue to the next page of the survey until 2 min had elapsed. Participants reported whether or not they had sent the message, then answered some questions about their emotions, and the barriers and motivators that they had considered when deciding whether or not to send their message.

**Measures.** Our pre-registered dependent variable was whether participants sent a message to their old friend, which we assessed the same way as in Studies 3 and 4. Participants also reported their current positive ($a$ = 0.87) and negative emotion ($a$ = 0.82), on the same scale as in Study 3.

We also asked participants several additional questions that are not reported in the main text, such as how much they considered the following while making their decision: a range of barriers, including worries about potential attributions that the target might make ($a$ = 0.77), and motivations ($a$ = 0.80), on the same measures as in Study 3 (see Supplementary Note, Study 7 for results related to these measures). We asked participants to describe the type of relationship they had with their old friend (81% were/had been close friends), how they knew their old friend (76% knew them from school), and how recently they had been in touch with their old friend ($M$ = 1.6, $SD$ = 0.9), using the same measures as in Study 6. We asked participants who had chosen to reach out to their old friend how glad they were to have sent the message ($M$ = 3.8, $SD$ = 0.8), and how glad they thought the recipient would be to have received the message ($M$ = 3.5, $SD$ = 1.0). Finally, we asked participants in the practice condition how many of their current friends/acquaintances that they had sent messages to during their practice session had responded before they decided whether or not to reach out to their old friend ($M$ = 1.1, $SD$ = 1.5).

### Reporting summary
Further information on research design is available in the Nature Portfolio Reporting Summary linked to this article.

## Results
### Study 1
A chi-square analysis revealed that a significant majority (90.9%) of participants had lost touch with a friend they care about, $X^2(1)$ = 295.5, $p$ < 0.001.

Yet, participants did not report being particularly willing to reach out to their old friend in the future, as evidenced by ratings ($M$ = 4.1, $SD$ = 1.9) that did not differ from the midpoint of the scale labeled as "neutral/undecided" according to a Bayesian one-sample test (assuming a diffuse distribution for priors on the variance and mean, and using a Monte Carlo approximation based on 10,000 samples), BF01 = 22.18 (strong evidence in favour of the null hypothesis), $t(400)$ = 0.52, $p$ = 0.60, $d$ = 0.03, $\Delta M$ = 0.05, $CI_{95}$ = [−0.14, 0.24]. Participants were even *less* willing to reach out to this same target right now ($M$ = 3.3, $SD$ = 2.0), with responses to this question falling significantly below the midpoint of the scale, $t(399)$ = −7.33, $p$ < 0.001, $d$ = −0.37, $\Delta M$ = −0.74, $CI_{95}$ = [−0.94, −0.54]. Both of these results hold after applying a Bonferroni correction for multiple comparisons. The hesitation to reach out is perplexing given that participants expected their friend to view them ($M$ = 4.3, $SD$ = 1.4) and their message ($M$ = 4.4, $SD$ = 1.4) positively (i.e., above the neutral scale midpoint), $t(400)$ = 4.79, two-tailed $p$ < 0.001, $d$ = .24, $\Delta M$ = 0.34, $CI_{95}$ = [0.20, 0.48], and $t(400)$ = 5.82, two-tailed $p$ < 0.001, $d$ = 0.29, $\Delta M$ = 0.40, $CI_{95}$ = [0.26, 0.53], respectively.

People indicated that a variety of barriers hold them back from reaching out (see Fig. 1). The most strongly endorsed barrier was a concern that the friend may not want to hear from them ($M$ = 5.2, $SD$ = 2.1), followed by a concern that it may be awkward to reach out after all this time ($M$ = 4.9, $SD$ = 2.2), both of which were endorsed above the midpoint of the scale using separate one-sample, non-directional t-tests, $t(400)$ = 11.05, $p$ < 0.001, $d$ = 0.55, $\Delta M$ = 1.16, $CI_{95}$ = [0.95, 1.37], and $t(400)$ = 7.63, $p$ < 0.001, $d$ = 0.38, $\Delta M$ = 0.85, $CI_{95}$ = [0.63, 1.06], respectively. Meanwhile, participants reported that only a few situations offered a legitimate reason for reaching out to their old friend. The most compelling reason for reaching out was their friend's birthday ($M$ = 4.8, $SD$ = 2.1), which was rated as significantly higher than the midpoint of the scale using a one-sample, non-directional t-test, $t(397)$ = 7.16, $p$ < 0.001, $d$ = 0.36, $\Delta M$ = 0.77, $CI_{95}$ = [0.56, 0.98] (see Fig. 2).

Study 1 revealed that the majority of people have lost touch with a friend they care about, but report neutral feelings, at best, about reaching out to their old friend. Further, people acknowledge that a wide range of barriers prevent them from reaching out and few reasons warrant them reaching out. These hesitations are notable in light of participants reporting that they expect themselves and their message to be well-received.

Does a reluctance to reach out to old friends stem from a hesitation to reconnect or a hesitation to initiate contact? Recent research suggests that people are particularly anxious about *initiating* conversations[36], so in Study 2 we examined whether one's willingness to reconnect differs depending on one's role in the exchange. Specifically, we predicted that people would be more willing to reconnect if their old friend initiated contact than if they were the one having to initiate.

### Study 2
Consistent with our pre-registered hypotheses, two independent samples t-tests found that participants were more interested in hearing from a friend, both now ($M$ = 4.9, $SD$ = 2.0) and in the future ($M$ = 5.4, $SD$ = 1.7), than reaching out to a friend ($M_{now}$ = 3.5, $SD$ = 1.9; $M_{future}$ = 4.8, $SD$ = 1.9; $t(197)$ = 4.93, one-tailed $p$ < 0.001, $d$ = 0.70, $\Delta M$ = 1.36, $CI_{95}$ = [0.82, 1.90], and $t(197)$ = 2.49, one-tailed $p$ = 0.01, $d$ = 0.35, $\Delta M$ = 0.63, $CI_{95}$ = [0.13, 1.14], respectively, see Fig. 3). This suggests that *initiating* contact may be a primary challenge to reconnecting.

Studies 1 and 2 demonstrate that people are surprisingly unwilling to reach out to an old friend, but self-reported responses may fail to capture how people actually behave. Therefore, in Study 3, we examined actual behaviour.

### Study 3
Across conditions, fewer than a third of participants (27.8%) reached out by sending a message to their old friend. We used a one-way ANOVA to test our pre-registered hypothesis that the proportion of people who sent their message would differ across conditions. Counter to predictions, we did not find evidence that reaching out rates differed across conditions, $F(2$,

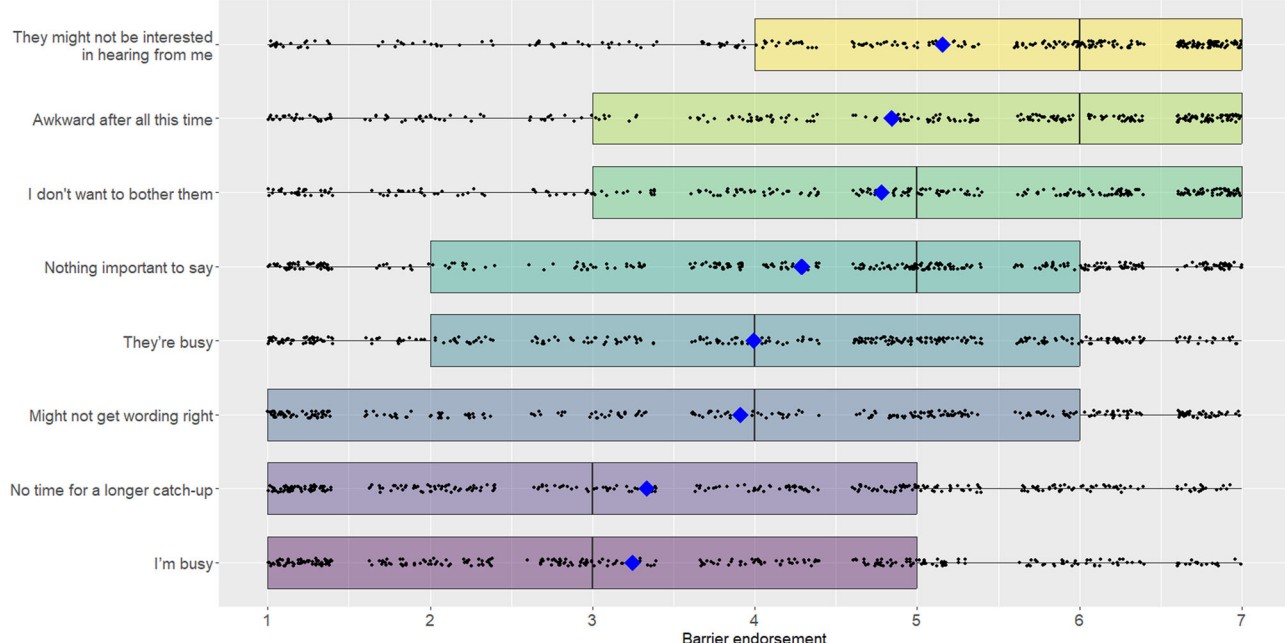

**Fig. 1 | Endorsement of various barriers to reaching out in Study 1.** Boxplot showing all the data; barring missing data, all participants ($N = 401$) rated all items. The upper and lower hinges of the boxplot correspond to the first and third quartiles (the 25th and 75th percentiles). The median is indicated by the line in the boxplot, and the mean is indicated by the blue diamond.

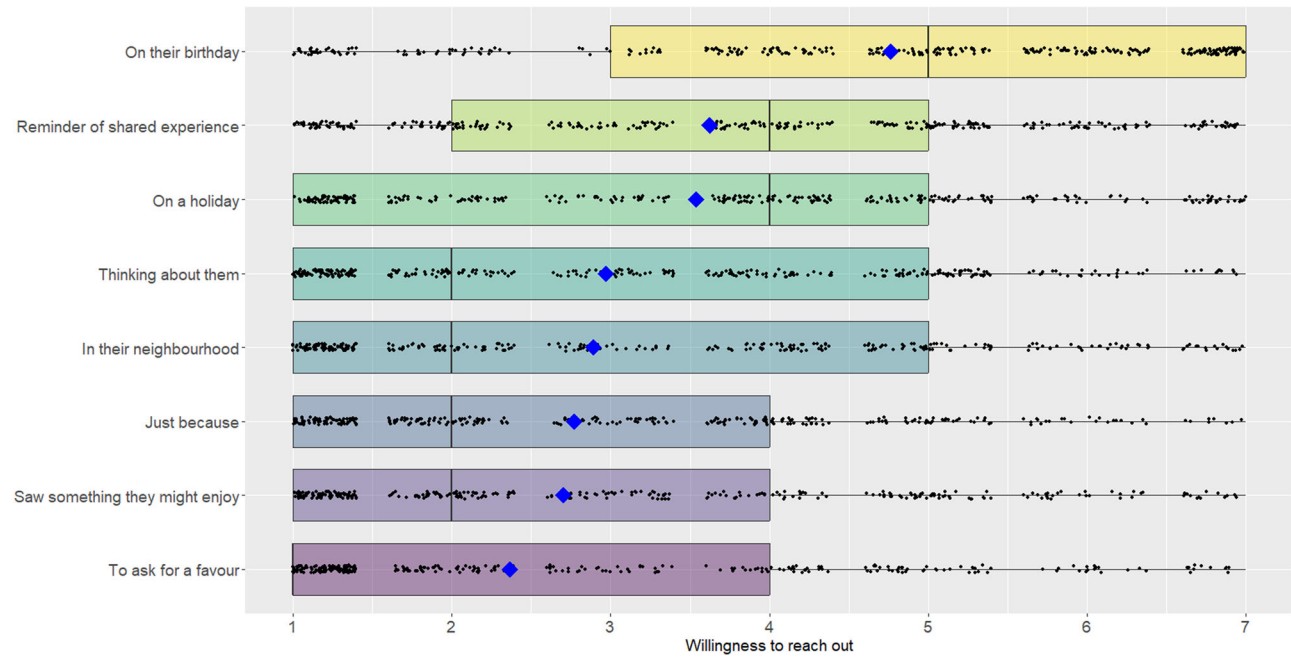

**Fig. 2 | Endorsement of various reasons for reaching out in Study 1.** Boxplot showing all the data; barring missing data, all participants ($N = 401$) rated all items. The upper and lower hinges of the boxplot correspond to the first and third quartiles (the 25th and 75th percentiles). The median is indicated by the line in the boxplot, and the mean is indicated by the blue diamond.

$450) = 1.72$, $p = 0.18$, $\eta_p^2 = 0.01$, $CI_{95} = [0.001, 0.03]$, with 27.5% sending the message in the control condition, 23.2% in the reflective condition, and 32.7% in the impulsive condition. Bayesian independent t-tests assuming unequal variance and using diffuse priors found moderate evidence in favour of the null hypothesis: $t(298) = 0.86$, $p = 0.39$, $BF01 = 7.65$ for reflective vs. control, and $t(300) = 0.98$, $p = 0.33$, $BF01 = 6.93$ for impulsive vs. control. Thus, we did not find evidence that encouraging people to adopt a reflective or impulsive thinking style increased the likelihood that they would reach out to an old friend.

Exploratory analyses indicated that participants who sent a message to their old friend reported more positive emotion ($M = 3.3$, $SD = 0.8$) and less negative emotion ($M = 1.5$, $SD = 0.6$) afterward than people who did not reach out ($M_{PA} = 2.6$, $SD = 0.9$; $M_{NA} = 1.7$, $SD = 0.7$), $t(451) = -7.22$, two-tailed $p < 0.001$, $d = 0.76$, $\Delta M = -0.64$, $CI_{95} = [-0.82, -0.47]$, and $t(451) = 2.96$, two-tailed $p = 0.002$, $d = 0.31$, $\Delta M = 0.22$, $CI_{95} = [0.07, 0.36]$, respectively. While these data are consistent with the idea that reaching out to an old friend is emotionally rewarding, the present data are correlational in nature and therefore cannot rule out the possibility that people

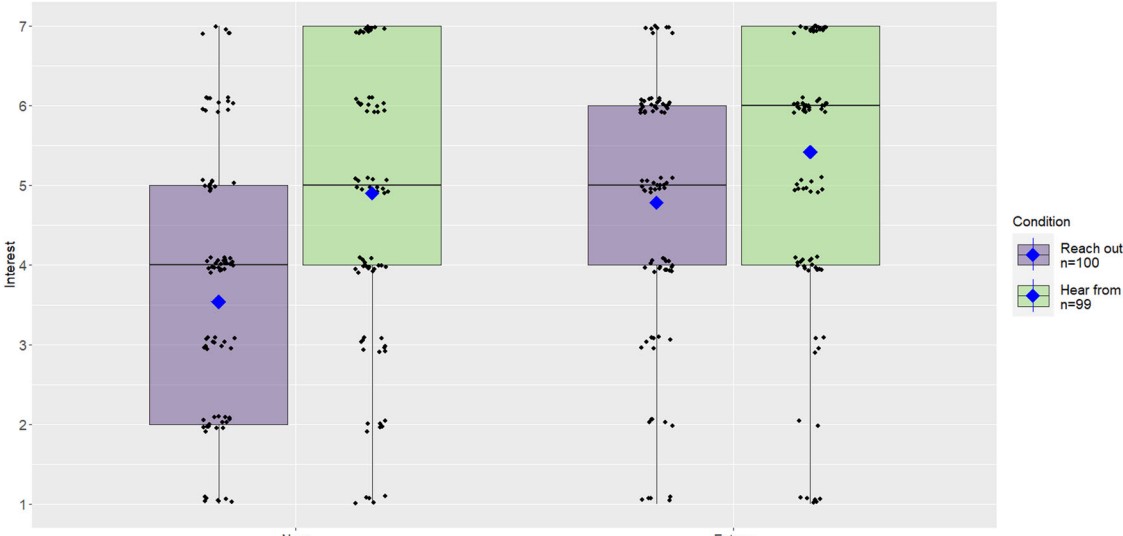

**Fig. 3 | Interest in reaching out to or hearing from an old friend now and in the future in Study 2.** Boxplot showing all the data; barring missing data, all participants (N = 199) rated their interest at both time points. The upper and lower hinges of the boxplot correspond to the first and third quartiles (the 25th and 75th percentiles). The median is indicated by the line in the boxplot, and the mean is indicated by the blue diamond.

experiencing greater positive emotions and lower negative emotions were more willing to reach out.

In Study 3, fewer than one third of people took the opportunity to reach out to an old friend, even though they wanted to reconnect with the target, thought the target wanted to hear from them, had the target's contact information, and were given time to draft and send a message. These findings converge with the self-reports from Studies 1–2 to further demonstrate that most people are reluctant to reach out to an old friend. In addition, the two interventions designed to encourage reaching out, by changing people's thinking about the act, were unsuccessful.

We wondered whether the low and relatively stable levels of reaching out in Study 3 may have been a result of the study design, therefore we made two changes in Study 4. First, we took a bottom-up approach to designing the intervention, targeting the particular barriers that participants endorsed in Studies 1–2 when thinking about reaching out to old friends. Additionally, it is possible that we did not detect differences across conditions in Study 3 because the control condition elevated reaching out rates by providing participants with time to write a message. Therefore, in Study 4, we designed a more realistic control condition.

## Study 4

We predicted that participants in both the *message* and *message plus encouragement* conditions would be more likely to reach out than participants in the control condition, and that people in the message plus encouragement condition would be more likely to reach out than participants in the message condition.

Across conditions, just over one third of participants (36.8%) reached out to a friend. Counter to predictions, a one-way ANOVA did not show evidence of different reaching out rates across conditions, $F(2, 601) = 1.93$, $p = 0.15$, $\eta_p^2 = 0.01$, $CI_{95} = [0.001, 0.02]$, with 42.3% of participants sending their message in the control condition, 33.3% in the message condition, and 35.0% in the message plus encouragement condition. Follow-up paired comparisons using a Tukey's test did not reveal any differences between the message and control conditions, $p = 0.16$, $\Delta M = 0.09$, $CI_{95} = [-0.02, 0.20]$, between the message plus encouragement and control conditions, $p = 0.28$, $\Delta M = 0.07$, $CI_{95} = [-0.04, 0.19]$, or between the two experimental conditions, $p = 0.94$, $\Delta M = -0.02$, $CI_{95} = [-0.13, 0.10]$. Similarly, Bayesian independent t-tests assuming unequal variance and using diffuse priors found some evidence in favour of the null hypothesis for message vs. control, $t(396) = -1.84$, $p = 0.07$, BF01 = 2.40 (anecdotal evidence), and for message

plus encouragement vs. control, $t(398) = -1.50$, $p = 0.13$, BF01 = 4.18 (moderate evidence). Thus, we did not find evidence to suggest that addressing people's concerns about reaching out increased the likelihood of reaching out to an old friend, and if anything, the interventions nudged participants in the *opposite* direction.

Studies 1–4 reveal that people both report and demonstrate a reluctance to reach out to old friends despite various forms of encouragement and the removal of several commonly cited barriers. This hesitation is problematic given that reaching out to old friends offers one meaningful route to social connection and, in turn, greater well-being. Where does this reluctance come from? Why are people unwilling to reach out to someone who they were once close to? One possibility is that old friends feel a lot like strangers, and therefore reaching out to an old friend might activate the same apprehensions that people have about talking to strangers.

A growing body of research demonstrates that people are unwilling to talk to strangers and avoid opportunities to do so. Indeed, despite several studies demonstrating that brief conversations with strangers can promote one's happiness and belonging[29,30], people report both avoiding and dreading these conversations due to a number of fears. For instance, people worry that they will not enjoy the conversation, not like their partner, and not have the necessary conversational skills (e.g., know how to start and maintain the conversation)[42]. In addition, people fear that their partner will not like them or enjoy the conversation[42]. Some of these common fears seem less relevant for old friends; people already know that they like the other person and presumably would only consider reaching out if they expected to enjoy the conversation. Indeed, in the present studies we specifically asked people to nominate an old friend that they would be happy to reconnect with. Yet, other fears seem more relevant. When reaching out to an old friend, people might worry that, even though they have interacted with the friend before, they will not know what to say after all this time, that their old friend may not be interested in hearing from them, and that the exchange will be awkward. Indeed, all of these concerns were endorsed to some degree in Study 1. Therefore, it seems plausible that people may harbour some of the same fears about reaching out to an old friend that they do when initiating a conversation with a stranger.

We explored the idea that old friends can feel like strangers in three remaining studies. Specifically, in Study 5 we examined the relative strength of people's reluctance to reach out to old friends by benchmarking the willingness to reconnect with an old friend against the willingness to talk to a stranger: an active, social, and commonly avoided behaviour[27,29]. Then, in

Study 6 we examined whether people are more reluctant to reach out to old friends when old friends feel more like strangers (i.e., whether familiarity acts as a mechanism). Finally, in Study 7, we applied lessons from one intervention shown to lastingly ease anxieties about talking to strangers. By assigning some participants to complete a warm-up activity, we effectively encouraged more people to reach out to old friends.

## Study 5

Participants' willingness ratings are shown in Fig. 4. On average, willingness to reach out to an old friend was lower than all but two of the seven other actions (book a medical appointment and sort a bag of coins), though the differences were not always statistically significant. Critically, as predicted, a Bayesian independent t-test assuming unequal variance, and using diffuse priors revealed that participants were no more willing to reach out to an old friend ($M = 4.6$, $SD = 1.7$) than they were to talk to a stranger ($M = 4.6$, $SD = 1.7$), $t(287) = -0.42$, $p = 0.67$, $d = 0.03$, BF01 = 19.52 (strong evidence in favour of the null hypothesis).

Again demonstrating people's reluctance to reach out to old friends, Study 5 revealed that people were no more willing to reach out to an old friend than they were to perform seemingly aversive activities, such as picking up litter or holding a handgrip for 30 s. Most notably, people were no more willing to reach out to an old friend than talk to a stranger, which raises an interesting possibility: people may be reluctant to reach out to old friends *because* they feel like strangers. In other words, one potential reason why people are unwilling to reach out to old friends is because old friends feel unfamiliar, like strangers. Therefore, we next examined whether people are less likely to reach out to old friends that feel more like strangers (and, conversely, more likely to reach out to old friends that feel more familiar), using a within-subjects design similar to past work[32].

## Study 6

Responses provided useful descriptive insight into the nature of old friendships. Specifically, participants indicated that old friends reflected various relationship types, including people who were, or had been, close friends (46%), social acquaintances (16%), family members (14%), and colleagues (13%; see Supplementary Fig. 5 in the SI for full descriptives). Old friends were primarily people that the participants knew from school (29%),

through friends/family (23%) or from work (22%; see Supplementary Fig. 6 in the SI for full descriptives). Most participants reported that they were last in touch with their old friend more than a few months ago ($M = 1.8$, $SD = 1.1$, Mode = 1), but 43% of participants had been in touch more recently.

Our primary, pre-registered hypothesis was that people would be less willing to reach out to old friends who feel less familiar. To test this hypothesis, we ran a linear mixed model using the lmer package in R[43], examining whether lower feelings of familiarity with an old friend predicted a lower willingness to reach out, with participant id entered as a random effect. As hypothesized, familiarity predicted willingness to reach out, $b = 0.63$, $SD = 0.03$, 95% CI = [0.58, 0.68], $t = 23.89$, indicating that people were less likely to reach out to old friends who felt less familiar (see Fig. 5). Of note, familiarity was also a significant predictor of reaching out to one's current friends, $r(319) = 0.51$, $p < 0.001$, and new acquaintances, $r(318) = 0.57$, $p < 0.001$.

Taken together, these results demonstrate that feelings of unfamiliarity toward an old friend predict a lower willingness to reach out. If reaching out to old friends can feel like talking to a stranger, can an intervention that reduces worries about talking to strangers encourage people to reach out?

Empirical evidence for the various benefits of talking to strangers is accumulating[29,30,44]. As a result, researchers have tested several strategies for encouraging people to talk to strangers more often. However, studies that have attempted to do so by reducing the fears that people have about talking to strangers have generally been unsuccessful[42]. One intervention, however, has been shown to lastingly change people's attitudes about talking to strangers. This intervention involves participants playing a scavenger hunt game in which they complete a "mission" every day for a week: talking to a stranger in the experimental condition, or observing a stranger in the control condition[45]. At the end of the week, participants in the experimental condition were less worried about rejection, and more confident in their ability to start and maintain a conversation. These changes in attitude persisted for at least a week after the intervention had ended. Importantly, this study found preliminary evidence that these changes in attitude might lead people to initiate more conversations with strangers.

Given the relative success of this design, we adapted it to our purposes here by asking participants in the experimental condition to complete a warm-up task in which they sent practice messages to current friends and

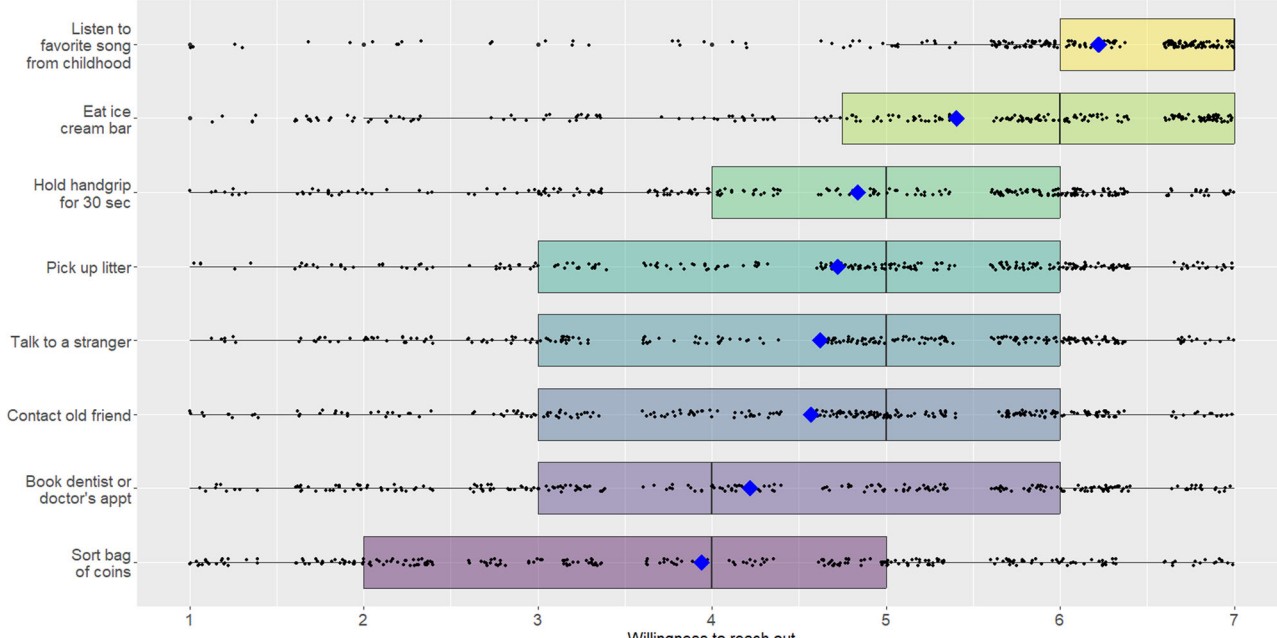

**Fig. 4 | Participant willingness to engage in eight common activities in Study 5.** Boxplot showing all the data; barring missing data, all participants ($N = 288$) rated all items. The upper and lower hinges of the boxplot correspond to the first and third quartiles (the 25th and 75th percentiles). The median is indicated by the line in the boxplot, and the mean is indicated by the blue diamond.

acquaintances. Meanwhile, participants in the control condition simply browsed social media: a similarly social, but more passive activity. We predicted that giving participants the opportunity to practice a form of the desired behaviour would encourage more people to reach out to old friends.

## Study 7

Consistent with our pre-registered prediction, more participants in the practice condition reached out to their old friend (53%) than did participants in the no-practice condition (31%), $t(194) = 3.20$, one-tailed $p < 0.001$, $d = -0.46$, $\Delta M = -0.22$, $CI_{95} = [-0.36, -0.09]$. Notably, the proportion of participants who reached out to an old friend in the control condition was (descriptively) similar to the proportions who reached out (across conditions) in our previous intervention studies: 27.8% in Study 3, and 36.8% in Study 4.

As in Study 3, people who reached out to their old friend reported more positive emotions ($M = 3.0$, $SD = .7$) than people who did not reach out ($M = 2.6$, $SD = 0.7$), $t(187) = 4.45$, one-tailed $p < 0.001$, $d = -0.65$, $\Delta M = -0.46$, $CI_{95} = [-0.66, -0.26]$, but unlike in Study 3, they did not differ in negative emotions, $t(187) = 1.26$, one-tailed $p = 0.10$, $d = 0.19$, $\Delta M = 0.12$, $CI_{95} = [-0.07, 0.30]$.

## Discussion

When friendships fade, are people eager and motivated to reach out and reconnect with old friends? Seven studies suggest that they are not. In Study 1, we saw that, although losing touch with a friend is an extremely common experience, most people express neutral or negative feelings about the prospect of reaching out to reconnect, citing several barriers and few reasons to do so. In Study 2, people were more willing to hear from vs. reach out to an old friend, which is consistent with the idea that people are especially hesitant about *initiating* contact, not about reconnecting. In Studies 3 and 4, we provided people with an opportunity to actually reach out to an old friend, and mitigated or removed several commonly cited barriers. Despite these aids, fewer than half of participants chose to reach out. Moreover, rates of reaching out were not meaningfully altered by a top-down manipulation informed by past research on dual processing models of human cognition (Study 3), nor a bottom-up manipulation that pre-emptively addressed

common concerns (Study 4), indicating that this tendency may be difficult to change.

After observing that people endorse similar fears when thinking about reaching out to an old friend as they do when thinking about talking to a stranger, we reasoned that one explanation for why people may avoid reaching out to old friends is that old friends feel like strangers after time has passed. To explore this possibility, in Study 5, we asked participants to rate their willingness to engage in several common daily tasks. We found that participants were no more willing to reach out to an old friend than they were to talk to a stranger. Moreover, in Study 6, we found that people were more reluctant to reach out to old friends when those friends felt more like strangers. Therefore, in Study 7, we adapted an intervention shown to ease anxieties about talking to strangers, which effectively increased by two-thirds the number of people who chose to reach out to an old friend.

The current findings add to the mounting body of research demonstrating that people undervalue social activities and actions[25]. Critically, this work also offers a number of extensions. First, we examine *behaviour* rather than (mis)predictions of how one thinks they would behave or how they expect themselves or others to feel. Indeed, Studies 3, 4, and 7 examine what proportion of participants actually reach out to old friends, which moves the literature beyond self-reports, expectations and misestimations, towards action[46]. Second, we document a reluctance to reach out to old friends in a range of relevant social contexts, such as being reminded of a shared memory or an upcoming holiday (Study 1), and in the face of several interventions (Studies 3-4). Thus, these data illustrate the pervasive nature of the reluctance to reach out. Finally, in Study 7, we provide evidence for an intervention that effectively increases reaching out to old friends - a behaviour that has informational and well-being benefits.

The intervention used in Study 7 to boost reaching out rates focused on changing peoples' behaviour by having them practice a version of the desired task. This intervention parallels the most successful strategy detected to date to encourage people to talk to strangers—simply practicing the task – and is a notable departure from most past research, which has tried and failed to promote social behaviour by educating or convincing people of the benefits of such actions[28,42]. Therefore, these findings align with recent theorizing on the potential benefits of targeting interventions toward the

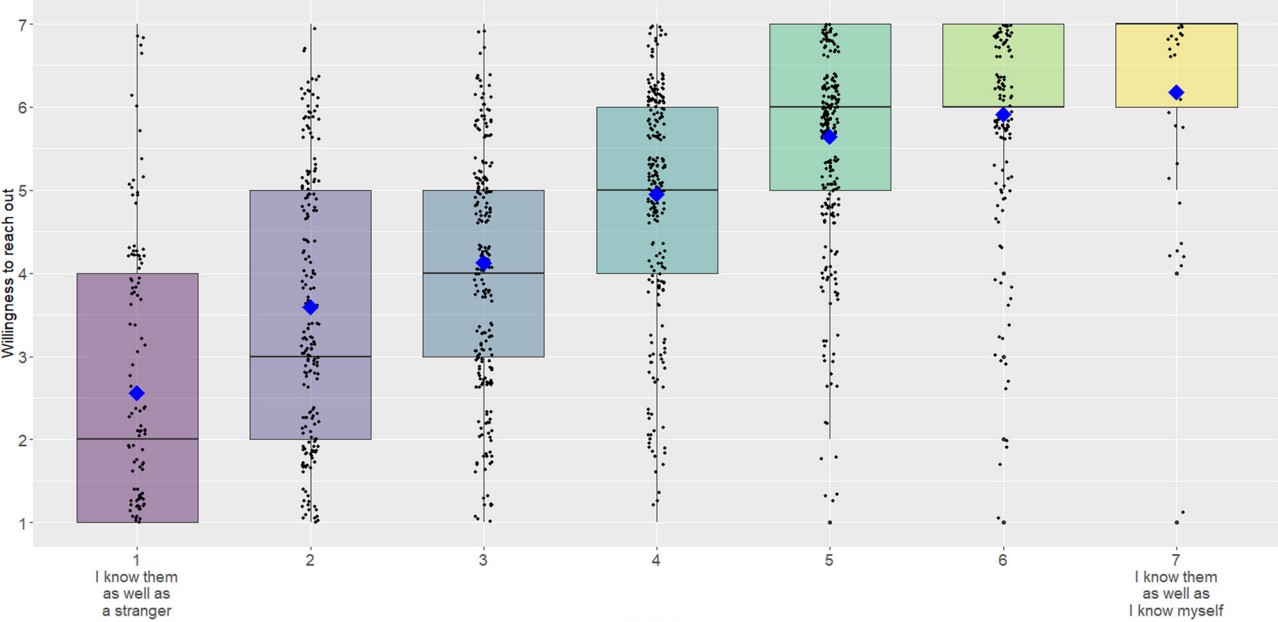

**Fig. 5 | Participant willingness to reach out to old friends who vary in familiarity in Study 6.** Boxplot showing all the data. Participants nominated 3 to 5 old friends who varied in familiarity: 1 = I know them as well as a stranger (N = 135), 2 (N = 215), 3 (N = 232), 4 (N = 254), 5 (N = 267), 6 (N = 150), 7 = I know them as well as I know myself (N = 45). The upper and lower hinges of the boxplot correspond to the first and third quartiles (the 25th and 75th percentiles). The median is indicated by the line in the boxplot, and the mean is indicated by the blue diamond.

social context or situation, and away from altering attitudes because the latter may be slower or more resistant to change[47].

Of course, this does not mean that peoples' attitudes and appreciation of the benefits of reaching out have no impact. Data from Study 1 revealed that the more participants thought their friend would appreciate them reaching out, the more willing they were to reach out to their friend now and in the future. Along similar lines, participants in Study 1 who saw reaching out as more of a prosocial act were more willing to engage in the behaviour, both now and in the future. These findings suggest that interventions designed to change peoples' minds or attitudes – by proactively signaling the recipient's appreciation or framing reaching out as an act of kindness—may ultimately be successful[28]. However, it is possible that these interventions must be more explicit or intensive to be effective because, by targeting attitudes, they are one step further removed from the behaviour they aim to change.

Reconnecting with old friends may bring opportunities for social connection and greater well-being, but this only happens if at least one party is willing to reach out. The present data suggest that people are generally interested in connecting, but prefer that the other person initiate (see Study 2). These findings align with previous work that finds that people are more interested in hearing personal information about others than they are in sharing similar information about themselves[48]. Is the hesitation to initiate because people assume that others are more likely to reach out than they truly are? In the SI, we report one study (Supplementary Note, Study S8) demonstrating that people overestimate the willingness of others to reach out. Specifically, participants read about the control condition in Study 3 and were asked to predict what percentage of participants would send a message to an old friend. Participants estimated that 56.6% would reach out, which was nearly double the actual percentage observed (27.5%). These data are consistent with the possibility that people think others will reach out, thereby relieving them of the task, and could be explored more deeply in the future. Indeed, Supplementary Note Study S9 in the SI demonstrates that people also overestimate their own willingness to reach out to old friends. Thus, people may hold various flawed assumptions about reaching out.

## Limitations

The present work has some limitations that can be considered in future research. First, the seven studies presented here considered reaching out to an old friend that participants *wanted* to reconnect with. Not all estranged friendships lapse from neglect; some friendships end on painful or angry terms, offering clear reason for disengagement. We focused on the former context both because we suspected this situation to be common, and because we thought it would provide a generous assessment of reaching out intentions and behaviour. Future researchers could consider how to encourage reaching out, if desirable, in more complicated relational contexts, such as when one or both parties are not eager.

Second, our studies collected data from participants in Western countries and the findings may therefore not generalize to other countries and contexts. Research on relational mobility suggests that in some contexts it is adaptive to have a wide network of weaker relationships, whereas in other contexts it is adaptive to maintain a smaller network of close relationships[49]. Future work could therefore expand this investigation to other cultural and socioeconomic contexts, which may differ in the extent to which they allow relationships to lapse, and value reconnecting when they do.

Finally, despite several studies examining people's willingness to reach out to an old friend and a stranger, we did not directly compare the experiences of these two actions. In light of past research and the present findings, we hypothesize that both experiences would be more positive than people expect. However, it is unclear which act would lead to greater momentary well-being. It seems plausible that reaching out to an old friend may promote greater happiness (than talking to a stranger) if the old friend responds quickly and positively, thus signaling mutual care in a way that is difficult to experience with strangers. This fascinating comparison remains an open question for future research.

## Implications

Western societies are growing increasingly concerned about loneliness and its dire impact on physical as well as psychological well-being[50]. Loneliness stems from perceiving fewer or lower quality social connections than one desires[51]. As a result, one intuitive idea about how to reduce loneliness is to help people build new social connections. However, building new social connections is difficult: it requires opportunities to meet new people, the social skills to initiate conversations with new people (i.e., strangers), not to mention repeated interaction and time spent together[22]. Alternatively, it might be easier and more efficient for people to revive existing relationships. Indeed, the empirically-informed Groups4Health program recommends just that[24]. However, the current research suggests that this recommendation may come with significant and previously unacknowledged challenges. The present findings suggest that more work is needed to understand how to break down the barriers, and support people in reaching out to reconnect.

Similarly, the current research suggests that a re-examination may be in order for one common positive psychology intervention for increasing well-being: practicing gratitude. People are often encouraged to write and send or deliver a thank you message to someone that they have not properly thanked. We suspect that, in practice, people often choose to thank someone who they have lost touch with: a favourite teacher from their school days or a workplace mentor from the early days of one's career. If this is the case, then people's predictions about what it will feel like to send the gratitude letter, and their decisions about whether or not to actually send the letter, are likely more complicated than formerly recognized; expressing gratitude may be confounded with reaching out to someone they have lost touch with. As a result, people may forgo opportunities to express gratitude, and ultimately experience greater happiness. Given the conceptual and practical overlap between reaching out and expressing gratitude, we hope researchers will investigate ways to help people overcome their hesitations to reach out, thereby making other happiness-boosting activities more likely as well.

Decades of research from across the social sciences indicates that relationships provide one of the most direct routes to happiness[1,52]. While recent years have expanded this examination to include brief interactions with strangers and acquaintances[29,30], the present work offers a timely and valuable reminder of one potentially overlooked source of social connection—reaching out to old friends. Indeed, we find that reaching out may also provide emotional benefits; participants in two of the present studies reported greater well-being after sending a message to an old friend than participants who opted not to do so (Studies 3 and 7). While the current data are correlational and should therefore be interpreted with caution, the observation that participants are happier after a social act is consistent with a large body of research demonstrating the hedonic rewards of brief social interactions and socialization[36,48,53]. Therefore, reaching out to old friends may offer an additional channel to social connection, and in turn, greater well-being.

## Conclusion

Relationships can fade for a variety of reasons. The present work demonstrates that the majority of people are reluctant to reach out to old friends, even when they are personally interested in doing so, believe their friend wants to hear from them, and are provided with time to draft and send a hello message. Moreover, this reluctance may be stubborn and difficult to change. One reason for this reluctance may be that old friends feel like strangers. Supporting this possibility, we find that people are no more willing to reach out to an old friend than they are to talk to a stranger, and that people are less willing to reach out to old friends who feel less familiar—more like strangers. Fortunately, one study reveals that people are more willing to reach out to an old friend after they practice the behaviour. More research is needed to understand how best to encourage people to reach out, so that they can experience the health and happiness benefits that come with increased social connection.

## Data availability

All materials and data are available on the Open Science Framework (OSF): https://osf.io/kydb3/.

## Code availability

Data were analyzed using SPSS 28.01 and R version 4.3.2. All code for analyses is available on the OSF: https://osf.io/kydb3/.

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

## Acknowledgements

We thank Marcel Aini, Anurada Amarasekera, Gurleen Bath, Lily Buttery, Kristina Castaneto, Dani Conception, Jaymie Cristobal, Katrina Del Villar, Fiona Eaket, Angie Fan, Amanda Hodges, Ravneet Hothi, Elyssa Hutchinson, Tori Kazemir, Allyson Klassen, Kalum Kumar, Erin Koch, Jacob Lauzon, Yassaman Malekzadeh, Katy Rogers, Marwan Saleh, Mia Sherley-Dale, Emily Stern, Naimah Sultana, Kelton Travis, Sophia Vennesland, and Rachael Whyte for their help with data collection, and Janaki Patel for her invaluable assistance with numerous tasks. The authors received no specific funding for this work.

## Author contributions

Lara B. Aknin developed the study concept, contributed to the study design, collected the data, analyzed the data, and drafted the manuscript. Gillian M. Sandstrom developed the study concept, contributed to the study design, collected the data, analyzed the data, and drafted the manuscript.

## Competing interests

The authors declare no competing interests.
