## [Peer Review File · Communications Psychology]

25th Aug 23

Dear Professor Aknin,

Thank you for your patience during the peer-review process. Your manuscript titled "How people decide when to inform others" has now been seen by 3 reviewers, and I include their comments at the end of this message. You will see that they find your work of some potential interest. However, they have raised quite substantial concerns that must be addressed. In light of these comments, we cannot accept the manuscript for publication, but would be interested in considering a revised version that fully addresses these serious concerns.

We hope you will find the Reviewers' comments useful as you decide how to proceed. Should additional work allow you to address these criticisms, we would be happy to look at a substantially revised manuscript. If you choose to take up this option, please highlight all changes in the manuscript text file, and provide a detailed point-by-point reply to the reviewers.

Editorially, we consider it important to conduct a new study or studies to address some of the concerns of Reviewers 1 and 3 regarding the psychological mechanisms for your effects. The reviewers have provided you with helpful suggestions that could help to illuminate why, for whom, or when we would expect the effect to occur. We also consider it important to make clearer the contribution of your work to the current question. We note that null effects for the current work are pre-registered using frequentist statistics and it would be important to ensure that these null effects have Bayesian evidence to support them. If you plan any additional studies that target null effects, they must be powered (at 80% or higher) to detect the smallest effect size of interest and base their conclusions on Bayesian statistics or equivalence tests.

For manuscripts that report null results, we require the following:

- Evidence that the study is sufficiently powered to detect the smallest theoretically or pragmatically meaningful effect
- Bayes Factors or equivalence tests to interpret the null results
- Appropriate language to describe the results. (There is no statistical test that can demonstrate absence of an effect. Statements such as 'There is no difference between x and y.' or 'X does not affect Y.' must be revised to read 'We found [no/little] credible evidence of a difference between x and y.' or 'We found [no/little] credible evidence that X affects Y.')

Please use the following link to submit your revised manuscript, point-by-point response to the referees' comments (which should be in a separate document to any cover letter) and the completed checklist:

[link redacted]

Please do not hesitate to contact me if you have any questions or would like to discuss these revisions further. We look forward to seeing the revised manuscript and thank you for the opportunity to review your work.

Best regards,

Patricia Lockwood

Patricia Lockwood, PhD
Editorial Board Member
Communications Psychology
orcid.org/0000-0001-7195-9559

EDITORIAL POLICIES AND FORMATTING

Editorial Policy: Policy requirements (Download the link to your computer as a PDF.)

Furthermore, please align your manuscript with our format requirements, which are summarized on the following checklist:

Communications Psychology formatting checklist

and also in our style and formatting guide Communications Psychology formatting guide .

* TRANSPARENT PEER REVIEW: Communications Psychology uses a transparent peer review system. This means that we publish the editorial decision letters including Reviewers' comments to the authors and the author rebuttal letters online as a supplementary peer review file. However, on author request, confidential information and data can be removed from the published reviewer

reports and rebuttal letters prior to publication. If your manuscript has been previously reviewed at another journal, those Reviewers' comments would not form part of the published peer review file.

* **CODE AVAILABILITY:** All Communications Psychology manuscripts must include a section titled "Code Availability" at the end of the methods section. In the event of publication, we require that the custom analysis code supporting your conclusions is made available in a publicly accessible repository; at publication, we ask you to choose a repository that provides a DOI for the code; the link to the repository and the DOI will need to be included in the Code Availability statement. Publication as Supplementary Information will not suffice. We ask you to prepare code at this stage, to avoid delays later on in the process.

* **DATA AVAILABILITY:**

All Communications Psychology manuscripts must include a section titled "Data Availability" at the end of the Methods section or main text (if no Methods). More information on this policy, is available at <http://www.nature.com/authors/policies/data/data-availability-statements-data-citations.pdf>.

At a minimum the Data availability statement must explain how the data can be obtained and whether there are any restrictions on data sharing. Communications Psychology strongly endorses open sharing of data. If you do make your data openly available, please include in the statement:

We recommend submitting the data to discipline-specific, community-recognized repositories, where possible and a list of recommended repositories is provided at <http://www.nature.com/sdata/policies/repositories>.

If a community resource is unavailable, data can be submitted to generalist repositories such as figshare or Dryad Digital Repository. Please provide a unique identifier for the data (for example a DOI or a permanent URL) in the data availability statement, if possible. If the repository does not provide identifiers, we encourage authors to supply the search terms that will return the data. For data that have been obtained from publicly available sources, please provide a URL and the specific data product name in the data availability statement. Data with a DOI should be further cited in the methods reference section.

REVIEWERS' EXPERTISE:

Reviewer #1 Social psychology, happiness, well-being
Reviewer #2 Social psychology, experiment, methods
Reviewer #3 Social psychology, experiment, methods

REVIEWERS' COMMENTS:

Reviewer #1 (Remarks to the Author):

This manuscript examines the extent to which people are willing to reach out to an old friend with whom they have lost touch, finding that they are surprisingly reluctant to reach out. This is an interesting and potentially important idea, and the paper is very well-written. I also appreciated the authors' commitment to evolving best practices in the field. That said, there may be some questions about the magnitude of the contribution here, especially given overlap with several other extant findings in the scientific literature (particularly Liu et al., 2022).

In Study 2, I found myself wondering whether activities like collecting litter were the right comparisons here. The one pre-registered hypothesis was that participants would be no more willing to reach out to an old friend than talk with a stranger, but this also made me curious about how much time had passed since each participant's last contact with the old friend they were considering. Indeed, an old friend could essentially be a relative stranger. If much time has elapsed, it's possible that people would know nearly nothing about someone else's life; for example, if participants thought of someone from their childhood, they would ostensibly not know anything about the adult they've become. Moreover, there is not clear evidence in this paper that actually reaching out to an old friend is any better than talking to a stranger. Would people be more interested in spending a day collecting litter and chatting with an old friend than spending the day collecting litter and chatting with a stranger? And would one of these in fact be a more positive interaction than the other? Tackling questions like these could be fruitful.

In Studies 3 and 4, it stood out to me that participants were only given two minutes to compose a message to their old friend. It seems plausible that participants did not have enough time to compose a message they were satisfied with, and this could be part of why some chose not to send it. It could also be the case that participants thought that this was not the right time to contact their old friend, for whatever reason, and that they preferred to send the message later. It wasn't obvious to me why the "maybe later" response was treated the same as a "no" response.

In Study 5, it's laudable that the researchers were able to access a nationally representative U.S. sample, but the benefits of collecting such data about people's beliefs are less clear when that sample's predictions are being directly compared to the convenience sample utilized in Study 3 (online panel participants from multiple countries, also including Canada and the U.K.). For studies like this, I think it typically makes sense to obtain expectations from the same population.

In my view, Study 6 was by far the most intriguing experiment in this empirical package. Much of the primary focus of this submission is about whether people reach out to old friends or not—some of the main results feel largely descriptive—but I believe the contribution would be more impactful if there was a greater emphasis on the underlying psychological explanations for why people can be surprisingly reluctant to reach out. The last study is the main one that gets at this, although even there one might question the novel contribution, as what is explored is similar to the 2022 Schroeder et al. publication (although, to be fair, that was in the context of interactions with strangers rather than with old friends). The investigation in the present research also reminded me of the work of Kardas et al. (2022) on people's hesitancy to engage in deeper conversations with others. There, participants were more interested in hearing meaningful information about someone else during a discussion than revealing information about themselves. The authors might benefit

from looking at that paper and thinking about some of the study designs in the context of reaching out to old friends. The current work had me thinking about Kumar and Epley's 2021 paper on voice-versus text-based interactions as well, as one of the experiments there was conducted in the context of reconnecting with old friends in particular.

I hope these researchers find this feedback helpful for their continued work, and I wish the authors the very best of luck in any further efforts.

Reviewer #2 (Remarks to the Author):

This is a creative, comprehensive, and well-controlled set of studies on an interesting and understudied topic that is important to people's every day lives. I recommend publication. Just a few tiny comments as follows:

Page 2 line 23. Typo: "most and reliable"

Page 5 line 105. How many t-tests, and was there correction for multiple comparisons? Was there adequate power/sample size to detect a difference from midpoint had there been one? "A number of t-tests" just reads as imprecise.

Page 8 line 149. What's the evidence that talking to a stranger is 'oft-dreaded' – is this a culture-specific finding? In any event, "oft" is pretty vague! Oft for whom? Oft compared to what? (Some relevant evidence is cited below so perhaps could also be cited up here).

Page 15 line 294. Typo: "message hello message"

Well done.

Reviewer #3 (Remarks to the Author):

This work explores people's willingness to reach out to old friends with whom one has lost touch. It provides several experiments that measure the willingness to reach out in general, measure the willingness to reach out after one has written up a message to an old friend, tests whether people overestimate the proportion of others willing to reach out, and tests whether people are more willing to hear from than to reach out to old friends.

Overall, the research topic is interesting and the data plentiful.

Having said this, it is not clear how the present experiments contribute to existing work (which the authors rightly cite) on people's apparent underestimation of the pleasantness/positivity of interacting with others and overestimation of the unpleasantness/negativity of reach out to old friends. It requires a more precise effort on the authors' part to explain how their experiments meaningfully contribute to that work. I am not saying that there aren't any contributions, but that

those should be more precisely and clearly stated.

Study 1 is a bit weak given that there is no meaningful comparison group – simply comparing the results to the midpoint of the scale is not particularly meaningful as a way to conclude that participants are unwilling to reach out to old friends.

Study 5 is not quite related to the present question because it asks people to predict others' (rather than their own) willingness to get in touch with old friends. Perhaps a more appropriate design would ask people to predict their own willingness to get in touch over a future defined time period, and then measure their accuracy in self-predictions after that time period elapses?

The purpose of Study 6 can be explained more precisely and clearly as it measures the willingness to reach out vs. the willingness to hear from – but if both old friends aren't willing to reach out then connection is less likely to be made.

The authors frame this work as helping people enhance well-being and happiness through social connections. In this vein, perhaps the most well-cited paper on this topic (Baumeister & Leary, 1995) suggests that it is close connections—and a small but dependable amount of them —that matter far and away the most for well-being and happiness rather than dormant/past friendships. Is it possible for someone to be perfectly happy interacting with current friends without reaching out to dormant/past friends? The authors' perspective on this will be most helpful for understanding the contribution of the present work. Is it that reconnecting with old friends adds a “cherry on top” to happiness, whereas connections with current friends is a hygiene feature that ensures that people are sufficiently happy (i.e. not experiencing depression/sadness)? Another perspective?

Baumeister, R. F., & Leary, M. R. (1995). The need to belong: Desire for interpersonal attachments as a fundamental human motivation. *Psychological Bulletin*, 117(3), 497–529.

Overall, the conclusion that people are reluctant to get in touch with past friends is more or less clear. However, the significance of this conclusion requires better explanation and discussion.

Smaller point

In Study 3, both treatment conditions probably tap into “System 2” as System 1 is supposed to be instinctive and often non-conscious decisions.

January 12, 2024

To the Three Anonymous Reviewers,

My collaborator, Dr. Gillian Sandstrom, and I would like to thank you for the thoughtful feedback and guidance on our manuscript, entitled "*People are Surprisingly Hesitant to Reach Out to Old Friends*" (COMMSPSYCHOL-23-0201-T). We have taken the feedback to heart and made some significant changes to our manuscript. We believe that the current version is stronger as a result, and are pleased to submit our revised manuscript to *Communication Psychology*.

Below we outline the changes that we have made to our manuscript in response to each reviewer's request. Our response to the Editor is in a separate cover letter, as requested by the journal.

Reviewer #1:

1. *This manuscript examines the extent to which people are willing to reach out to an old friend with whom they have lost touch, finding that they are surprisingly reluctant to reach out. This is an interesting and potentially important idea, and the paper is very well-written. I also appreciated the authors' commitment to evolving best practices in the field.*

We thank Reviewer 1 for their supportive comments regarding the value, clarity, and rigor of the manuscript.

2. *That said, there may be some questions about the magnitude of the contribution here, especially given overlap with several other extant findings in the scientific literature (particularly Liu et al., 2022).*

We appreciate Reviewer 1's encouragement to identify the unique contribution of our work and have taken the opportunity to do so in the revised manuscript. As suggested, we now explicitly identify our contribution in the introduction (pages 5-6):

Critically, systematically underestimating others' appreciation for one's social behaviour (e.g., reaching out, talking to a stranger, giving a compliment) is expected to make people more reluctant to engage in these activities because they are missing the full motivation to act. While this premise is grounded in common sense and theory (Becker, 1993), past work has rarely examined whether and to what extent people are willing to engage in the relevant behaviour. The only such investigation we are aware of was conducted by Liu and colleagues (2022; Study S4 in SOM) who attempted to increase the number of people reaching out to an old friend by teaching participants about misestimation errors.

Unfortunately, this intervention did not translate into more people actually reaching out to an old friend. Thus, while past work has demonstrated that people systematically underappreciate how much social targets value interactions, including being contacted by an old friend, research has yet to explore people's self-reported and actual willingness to engage in these actions, as well as how to promote this valuable behaviour.

And in the general discussion (pages 24-25):

The current findings add to the mounting body of research demonstrating that people undervalue social activities and actions (Kumar & Epley, 2023). Critically, this work also offers a number of extensions. First, we examine behaviour rather than (mis)predictions of how one thinks they would behave or how they expect themselves or others to feel. Indeed, Studies 3, 4, and 7 examine what proportion of participants actually reach out to old friends, which moves the literature beyond self-reports, expectations and misestimations, towards action (Baumeister, Vohs & Funder, 2007). Second, we document a reluctance to reach out to old friends in a range of relevant social contexts, such as being reminded of a shared memory or an upcoming holiday (Study 1), and in the face of several interventions (Studies 3-4). Thus, these data illustrate the pervasive nature of the reluctance to reach out. Finally, in Study 7, we provide evidence for an intervention that effectively increases reaching out towards old friends. While only one study that we are aware of has attempted to encourage this behaviour in the past (Liu et al, 2022; Study S4), this study did not successfully alter behaviour. Thus, the data reported in Study 7 is the first to successfully increase the proportion of people who reach out to old friends - a behaviour that has informational and well-being benefits.

- 3. In Study 2, I found myself wondering whether activities like collecting litter were the right comparisons here. The one pre-registered hypothesis was that participants would be no more willing to reach out to an old friend than talk with a stranger, but this also made me curious about how much time had passed since each participant's last contact with the old friend they were considering. Indeed, an old friend could essentially be a relative stranger. If much time has elapsed, it's possible that people would know nearly nothing about someone else's life; for example, if participants thought of someone from their childhood, they would ostensibly not know anything about the adult they've become. Moreover, there is not clear evidence in this paper that actually reaching out to an old friend is any better than talking to a stranger. Would people be more interested in spending a day collecting litter and chatting with an old friend than spending the day collecting litter and chatting with a stranger? And would one of these in fact be a*

more positive interaction than the other? Tackling questions like these could be fruitful.

We thank Reviewer 1 for their feedback. In response to these queries we have made the following edits:

- In the benchmarking study (Study 2 in the original submission, Study 5 in the revised submission), we now explain how the comparison behaviours were selected. Specifically, we now note on page 17 that the comparison behaviours were selected to represent a range of positive, neutral and negative actions from everyday life.
- We deeply appreciate Reviewer 1's curiosities regarding when participants were last in touch with an old friend, whether old friends can sometimes feel like strangers, and if a sense of familiarity (or lack thereof) predicts one's willingness to reach out to an old friend. These insightful questions motivated Study 6 in the revised manuscript wherein we ask participants to identify 3-5 old friends and, for each individual indicate: (i) the type of relationship (e.g., friend, colleague, neighbor), (ii) how long ago they last interacted, (iii) how well they know this person now (like a stranger vs. as well as I know myself), and (iv) how willing they are to reach out to this person. The new data not only provide insight into the recency of last contact, which most participants rate as being over a few months ago, but they also critically show that people are less likely to reach out to an old friend when they feel more like a stranger. Importantly, we agree with Reviewer 1 that these data provide a key insight into one potential mechanism for why people are reluctant to reach out to old friends – old friends can feel like strangers – which we have made a central component of the revised manuscript.

4. *In Studies 3 and 4, it stood out to me that participants were only given two minutes to compose a message to their old friend. It seems plausible that participants did not have enough time to compose a message they were satisfied with, and this could be part of why some chose not to send it. It could also be the case that participants thought that this was not the right time to contact their old friend, for whatever reason, and that they preferred to send the message later. It wasn't obvious to me why the "maybe later" response was treated the same as a "no" response.*

We understand how 2-minutes may seem like a short window of time for participants to craft a message that they are comfortable sending to an old friend.

However, we argue that this seems like an unlikely explanation for the reluctance to reach out to old friends for at least three reasons.

1. First, the average willingness to reach out to an old friend in Studies 1-2 is neutral or negative, even when participants are simply asked to think about reaching out to an old friend without any time limit being mentioned.
2. Second, participants in Study S9 (located in the SOM) were given *one full week* to reach out to as many old friends as they could or desired and were sent a reminder to do so. Even with this scaffolding and freedom to draft a message of their choice at their leisure, participants only reported reaching out to an average of 2-3 old friends, which was significantly lower than their targeted number (an average of 8.6 old friends).
3. Finally, over half of participants (52%) in Study 7, who were also given a two-minute limit, were able to send a message to an old friend in the experimental condition that provided participants with a “warm up” activity.

If you would like us to directly address this point in the manuscript, please let us know and we would be willing to do so.

Participants were asked to indicate whether they had sent a message to their old friend by selecting one of three response options: “yes”, “no” or, “maybe later.” Given past research demonstrating that participants can be eager to please researchers (perhaps especially so when payment is pending; Brito, 2017; Nicholas & Maner, 2008), we provided the response option “maybe later” to increase honest responding. We figured that by providing participants with another socially desirable and acceptable response, this would reduce the pressure for participants to say “yes” they had sent their message when they had not yet done so. Finally, we wish to reiterate that the decision to treat “maybe later” responses like “no” responses was pre-registered in advance to parallel our focus on behaviour (rather than intentions) and appears appropriate given that most intentions to reach out are not enacted (Study S9).

5. *In Study 5, it's laudable that the researchers were able to access a nationally representative U.S. sample, but the benefits of collecting such data about people's beliefs are less clear when that sample's predictions are being directly compared to the convenience sample utilized in Study 3 (online panel participants from multiple countries, also including Canada and the U.K.). For studies like this, I think it typically makes sense to obtain expectations from the same population.*

In light of this comment, we have moved the national panel prediction study (Study 5 in the original submission) to the Supporting Online Materials (SOM);

now Study S8). This new location means the findings of this investigation receive less attention, which may be warranted given that we asked a nationally representative panel to predict the reaching out behaviours of a non-representative online panel. We do note, however, that Study S9 (presented in the SOM) provides what we believe to be a helpful extension by demonstrating that people also underestimate *their own* reaching out behaviours.

- In my view, Study 6 was by far the most intriguing experiment in this empirical package. Much of the primary focus of this submission is about whether people reach out to old friends or not—some of the main results feel largely descriptive—but I believe the contribution would be more impactful if there was a greater emphasis on the underlying psychological explanations for why people can be surprisingly reluctant to reach out. The last study is the main one that gets at this, although even there one might question the novel contribution, as what is explored is similar to the 2022 Schroeder et al. publication (although, to be fair, that was in the context of interactions with strangers rather than with old friends). The investigation in the present research also reminded me of the work of Kardas et al. (2022) on people’s hesitancy to engage in deeper conversations with others. There, participants were more interested in hearing meaningful information about someone else during a discussion than revealing information about themselves. The authors might benefit from looking at that paper and thinking about some of the study designs in the context of reaching out to old friends. The current work had me thinking about Kumar and Epley’s 2021 paper on voice versus text-based interactions as well, as one of the experiments there was conducted in the context of reconnecting with old friends in particular.*

We thank Reviewer 1 for their helpful encouragement to consider potential mechanisms behind a reluctance to reach out to old friends and look to previous work by Kumar and Epley (2021) for study design inspiration. We are pleased to say that we have done both in this revision. Specifically, we now propose that one reason people may be reluctant to reach out to old friends is because old friends can feel like strangers. To probe this possibility, we conducted Study 6 to investigate whether people are less willing to reach out to old friends who feel less familiar/more like a stranger. Data are consistent with this pre-registered prediction. Of note, this study adopted a within-subjects design, which is similar to Study 3b in Kumar and Epley’s (2018) paper examining mispredictions around expressing gratitude. We are very grateful for the suggestion!

7. *I hope these researchers find this feedback helpful for their continued work, and I wish the authors the very best of luck in any further efforts.*

We thank Reviewer 1 for their very helpful feedback, which was critical in revising this manuscript.

Reviewer #2:

1. *This is a creative, comprehensive, and well-controlled set of studies on an interesting and understudied topic that is important to people's every day lives. I recommend publication. Just a few tiny comments as follows:*

We thank Reviewer 2 for their kind feedback and support of this work!

2. *Page 2 line 23. Typo: "most and reliable"*

We thank Reviewer 2 for their close read of the manuscript and for catching this typo. The text has been revised to "most reliable."

3. *Page 5 line 105. How many t-tests, and was there correction for multiple comparisons? Was there adequate power/sample size to detect a difference from midpoint had there been one? "A number of t-tests" just reads as imprecise.*

We agree that the original wording was vague and so we have removed the imprecise language about "A number of t-tests." On pages 7-8 of the revised draft, we now specify that we conducted two separate tests examining whether people are willing to reach out to an old friend "now" and "in the future." In response to the Editor's request, the first test is now a one-sample Bayesian t-test, which offers strong evidence in favour of the null hypothesis, suggesting that people's willingness to reach out to old friends in the future does not differ from a neutral rating. The second test is similar, comparing willingness ratings to reach out to an old friend *now* to the neutral midpoint of the scale. This, however, did not require Bayesian analysis, because the effect is not null. We find that average willingness ratings are significantly below the midpoint, even after applying a Bonferroni correction to account for the fact that there are two comparisons.

4. *Page 8 line 149. What's the evidence that talking to a stranger is 'oft-dreaded' – is this a culture-specific finding? In any event, "oft" is pretty vague! Oft for whom?*

Oft compared to what? (Some relevant evidence is cited below so perhaps could also be cited up here).

We appreciate the opportunity to clarify our wording. We have revised the statement in question to say *“How does the reluctance to reach out to an old friend compare to people’s reluctance to do other things, including talk to strangers, an active, social, and commonly avoided (Epley & Schroeder, 2014; Kumar & Epley, 2022)?”* This new wording more accurately captures past research in which other authors have argued that people “sometimes” or “routinely” avoid talking to strangers (Epley & Schroeder, 2014). As suggested, we have included citations to two relevant papers following the revised sentence to properly support this claim.

5. *Page 15 line 294. Typo: “message hello message”*

Again, we thank Reviewer 2 for their close read of the manuscript and for catching this typo. The revised manuscript does not include the sentence in question, so the typo has been removed.

6. *Well done.*

Once again, we appreciate Reviewer 2’s positive evaluation of the research.

Reviewer #3:

1. *This work explores people’s willingness to reach out to old friends with whom one has lost touch. It provides several experiments that measure the willingness to reach out in general, measure the willingness to reach out after one has written up a message to an old friend, tests whether people overestimate the proportion of others willing to reach out, and tests whether people are more willing to hear from than to reach out to old friends. Overall, the research topic is interesting and the data plentiful.*

We are pleased to see that Reviewer 3 found the topic of investigation interesting and the data to be abundant.

2. *Having said this, it is not clear how the present experiments contribute to existing work (which the authors rightly cite) on people’s apparent underestimation of the pleasantness/positivity of interacting with others and overestimation of the unpleasantness/negativity of reach out to old friends. It requires a more precise*

effort on the authors' part to explain how their experiments meaningfully contribute to that work. I am not saying that there aren't any contributions, but that those should be more precisely and clearly stated.

We thank Reviewer 3 (like Reviewer 1 and the Editor) for encouraging us to clarify how the present research extends the literature meaningfully beyond what has been demonstrated before. As noted in our response to Reviewer 1 (point 2), we think this work offers at least three critical extensions by (i) going beyond self-reports to examine actual reaching out behaviour, (ii) demonstrating that the reluctance to reach out to old friends is reported in a range of social contexts, and (iii) providing evidence for one intervention that encourages more people to reach out to old friends, which, to our knowledge, has not been demonstrated previously.

3. *Study 1 is a bit weak given that there is no meaningful comparison group – simply comparing the results to the midpoint of the scale is not particularly meaningful as a way to conclude that participants are unwilling to reach out to old friends.*

We agree that Study 1 is not the most persuasive evidence for a general reluctance to reach out to old friends in isolation. However, we view Study 1 as an initial investigation or first look, which is supported by subsequent studies with converging findings that include comparison groups (Study 2), behavioural outcomes (e.g., Studies 3-4), and comparison activities (Study 5).

4. *Study 5 is not quite related to the present question because it asks people to predict others' (rather than their own) willingness to get in touch with old friends. Perhaps a more appropriate design would ask people to predict their own willingness to get in touch over a future defined time period, and then measure their accuracy in self-predictions after that time period elapses?*

We agree with Reviewer 3 in noting that Study 5 in the original manuscript was less relevant to the central question of investigation. As such, we have moved the study to the SOM where it is now Study S8 in the revised manuscript. This new location means the study receives less attention and does not distract potential readers from the focus of the paper.

That said, we found Reviewer 3's suggestion to consider whether people mispredict *their own* willingness to reach out to old friends to be deeply interesting. As a result, we conducted Study S9 in which we told participants

about the benefits of reaching out to old friends and asked them to indicate how many old friends they could commit to reaching out to in the upcoming week. Three days later, participants were sent a reminder to reach out to old friends. Finally, one week after the initial survey, participants were contacted to report how many old friends they reached out to. Consistent with our pre-registered predictions, participants reached out to significantly fewer old friends than they planned. We thank Reviewer 3 for inspiring this additional investigation.

5. *The purpose of Study 6 can be explained more precisely and clearly as it measures the willingness to reach out vs. the willingness to hear from – but if both old friends aren't willing to reach out then connection is less likely to be made.*

We appreciate the suggestion to present the results of Study 6 (now Study 2) more precisely by specifying that we measured the extent to which people were interested in reaching out vs. hearing from an old friend.

We also agree with Review 3's suggestion that "if both old friends aren't willing to reach out then the connection is less likely to be made." As noted in the general discussion (page 26), we started here by exploring contexts in which one person *wants* to reconnect with an old friend (and thinks the other person wants to hear from them). However, this may not always be the case. Thus, future research could explore if and how to reach out to old friends in more complicated relational contexts.

6. *The authors frame this work as helping people enhance well-being and happiness through social connections. In this vein, perhaps the most well-cited paper on this topic (Baumeister & Leary, 1995) suggests that it is close connections—and a small but dependable amount of them—that matter far and away the most for well-being and happiness rather than dormant/past friendships. Is it possible for someone to be perfectly happy interacting with current friends without reaching out to dormant/past friends? The authors' perspective on this will be most helpful for understanding the contribution of the present work. Is it that reconnecting with old friends adds a "cherry on top" to happiness, whereas connections with current friends is a hygiene feature that ensures that people are sufficiently happy (i.e. not experiencing depression/sadness)?*

We agree that classic work has demonstrated that having a small but dependable number of close connections is vital to well-being (Baumeister &

Leary, 1995; Diener & Seligman, 2002). However, in the revised introduction we now briefly summarize recent evidence suggesting that larger and more diversified social networks are associated with greater well-being, beyond close connections. Therefore, while people may be happy with their current friendships, they may be happier if they were to reach out to old friends. On page 3 we state:

While the quality of relationships matters, so too does the quantity and diversity of social connections. Social network size is positively associated with greater well-being (Rafnsson et al., 2015; Wang, 2016) and recent work spanning multiple international data sets indicates that people who have more diverse relationship networks also report greater well-being (Collins et al., 2022). These findings align with recent theorizing in relationship science which cautions against relying on any one person to fulfill all of one's emotional needs (Finkel et al., 2015). Instead, people who turn to different social connections for different emotion regulation needs (e.g., calling on one person to cheer them up when they are sad, and a different person to calm them down when they are anxious) report higher well-being (Cheung et al., 2015). Thus, although classic work indicates that high-quality relationships are necessary for happiness (Baumeister & Leary, 2005; Diener & Seligman, 2002), recent research suggests that having more diverse relationships is also a predictor of well-being.

7. *Overall, the conclusion that people are reluctant to get in touch with past friends is more or less clear. However, the significance of this conclusion requires better explanation and discussion.*

We thank Reviewer 3 for the opportunity to highlight the significance of this work. In response to this suggestion we have added and/or revised three paragraphs in an “implications” section of the general discussion (pages 27-29).

8. *Smaller point: In Study 3, both treatment conditions probably tap into “System 2” as System 1 is supposed to be instinctive and often non-conscious decisions.*

We agree that both treatment conditions in Study 3 may include a degree of cognitive oversight and therefore tap in “System 2,” albeit to potentially different degrees. As a result, we have adjusted the presentation of Study 3. We now explain that the two interventions reflect two common routes of cognition: a more effortful route vs. more immediate/reactive route. We think this revised wording helps to avoid misleading or confusing potential readers, while also tying the present work to the broader decision-making literature.

Again, we thank you for the constructive feedback on our manuscript. If you feel that we have misunderstood or inadequately addressed any of the comments above, please do not hesitate to let us know.

Sincerely,

Lara Aknin and Gillian Sandstrom

References

- Brito, C. F. (2017). Demonstrating experimenter and participant bias. In J. R. Stowell & W. E. Addison (Eds.), *Activities for teaching statistics and research methods: A guide for psychology instructors* (pp. 94–97). American Psychological Association.
<https://doi.org/10.1037/0000024-020>
- Nichols, A. L., & Maner, J. K. (2008). The good-subject effect: investigating participant demand characteristics. *The Journal of General Psychology*, *135*(2), 151–165.
<https://doi.org/10.3200/GENP.135.2.151-166>

8th Feb 24

Dear Professor Aknin,

Your manuscript titled "People are Surprisingly Hesitant to Reach Out to Old Friends" has now been seen by our reviewers, whose comments appear below. In light of their advice I am delighted to say that we are happy, in principle, to publish a suitably revised version in Communications Psychology under the open access CC BY license (Creative Commons Attribution v4.0 International License).

We therefore invite you to revise your paper one last time to address the remaining concerns of our reviewers and a list of editorial requests. At the same time we ask that you edit your manuscript to comply with our format requirements and to maximise the accessibility and therefore the impact of your work.

EDITORIAL REQUESTS:

SUBMISSION INFORMATION:

OPEN ACCESS:

Communications Psychology is a fully open access journal. Articles are made freely accessible on publication under a CC BY license (Creative Commons Attribution 4.0 International License). This license allows maximum dissemination and re-use of open access materials and is preferred by many research funding bodies.

For further information about article processing charges, open access funding, and advice and support from Nature Research, please visit <https://www.nature.com/commspsychol/article-processing-charges>

At acceptance, you will be provided with instructions for completing this CC BY license on behalf of all authors. This grants us the necessary permissions to publish your paper. Additionally, you will be asked to declare that all required third party permissions have been obtained, and to provide billing information in order to pay the article-processing charge (APC).

* TRANSPARENT PEER REVIEW: Communications Psychology uses a transparent peer review system. On author request, confidential information and data can be removed from the published reviewer

reports and rebuttal letters prior to publication. If you are concerned about the release of confidential data, please let us know specifically what information you would like to have removed. Please note that we cannot incorporate redactions for any other reasons.

* CODE AVAILABILITY: All Communications Psychology manuscripts must include a section titled "Code Availability" at the end of the methods section. We require that the custom analysis code supporting your conclusions is made available in a publicly accessible repository at this stage; please choose a repository that generates a digital object identifier (DOI) for the code; the link to the repository and the DOI must be included in the Code Availability statement. Publication as Supplementary Information will not suffice.

* DATA AVAILABILITY:

[link redacted]

Best regards,

Jennifer Bellingtier

Jennifer Bellingtier, PhD
Senior Editor
Communications Psychology

and on behalf of

Patricia Lockwood, PhD
Editorial Board Member
Communications Psychology
orcid.org/0000-0001-7195-9559

REVIEWERS' EXPERTISE:

Reviewer #1 Social psychology, happiness, well-being

Reviewer #2 Social psychology, experiment, methods

Reviewer #3 Social psychology, experiment, methods

REVIEWERS' COMMENTS:

Reviewer #1 (Remarks to the Author):

I continue to think this manuscript is in an interesting area of inquiry and appreciate the work these researchers put into a revision. Indeed, the authors deserve praise for being particularly responsive to the comments they received. I am especially glad that my feedback inspired their updated theorizing and new data collection efforts (i.e., the added Study 6 in the revised version).

In order to make for a more impactful contribution, I am still curious about whether these researchers have thoughts—or, even better, data—on how the experience of reaching out to an old friend actually compares to talking to a stranger. Is one likely to be more positive than the other? One limitation that remains here is that although the studies included in this empirical package examine willingness to reach out to an old friend versus talk to a stranger, they do not compare actual experiences of reaching out to an old friend versus talking to a stranger.

I wish the authors much luck as they pursue further research on this fascinating topic.

Reviewer #2 (No further remarks)

Reviewer #3 (Remarks to the Author):

I accept the ways the authors addressed the points made in the previous submission and so I do not have further critical comments for the authors. I congratulate the authors for an interesting paper.

March 4, 2024

Dear Dr. Lockwood,

My collaborator, Dr. Gillian Sandstrom, and I would like to thank you and the three reviewers for the conditional acceptance of our manuscript, entitled “*People are Surprisingly Hesitant to Reach Out to Old Friends*” (COMMSPSYCHOL-23-0201A). We are delighted to address the final substantial point of feedback and share a revised draft meeting the formatting requirements for publication at *Communication Psychology*.

Below we respond to the single remaining request and thank the three reviewers.

REVIEWERS' COMMENTS:

Reviewer #1:

1. *I continue to think this manuscript is in an interesting area of inquiry and appreciate the work these researchers put into a revision. Indeed, the authors deserve praise for being particularly responsive to the comments they received. I am especially glad that my feedback inspired their updated theorizing and new data collection efforts (i.e., the added Study 6 in the revised version).*

In order to make for a more impactful contribution, I am still curious about whether these researchers have thoughts—or, even better, data—on how the experience of reaching out to an old friend actually compares to talking to a stranger. Is one likely to be more positive than the other? One limitation that remains here is that although the studies included in this empirical package examine willingness to reach out to an old friend versus talk to a stranger, they do not compare actual experiences of reaching out to an old friend versus talking to a stranger.

We thank Reviewer 1 for their sustained interest in this work and kind words on the responsiveness of our revision.

As suggested, we have added a paragraph to the General Discussion sharing our thoughts on how the experiences of reaching out to an old friend and talking to stranger may compare. Specifically, on page 32 of the revised draft we say:

“Finally, despite several studies examining people's willingness to reach out to an old friend and a stranger, we did not directly compare the experiences of these two actions. In light of past research and the present findings, we hypothesize that both experiences would be more positive than people expect. However, it is unclear which act would lead to greater momentary well-being. It seems plausible that reaching out to an old friend may promote greater happiness (than talking to a stranger) if the old friend responds quickly and positively, thus signaling mutual care in a way that is difficult to experience with strangers. This fascinating comparison remains an open question for future research.”

Reviewer #2:

No comments provided. Thank you for your time and feedback.

Reviewer #3:

1. *I accept the ways the authors addressed the points made in the previous submission and so I do not have further critical comments for the authors. I congratulate the authors for an interesting paper.*

Thank you for your time, feedback, and congratulatory remarks.

Again, we thank you and the three reviewers for the constructive feedback and support of this work.

Sincerely,

Lara Aknin and Gillian Sandstrom